# Socioeconomic status and older adult's experiences of weight loss: a qualitative secondary analysis

Anna Newton-Clarke[1]*, Miriam J. Johnson[2], Ugochinyere Nwulu[3], Fliss E.M. Murtagh[4], Alex F. Bullock[5]

1 Junior Clinical Fellow, Yorkshire and Humber Health Education England, Wolfson Palliative Care Research Centre, Hull York Medical School, University of Hull, Hull, 2 Professor of Palliative Medicine, Wolfson Palliative Care Research Centre, Hull York Medical School, University of Hull, Hull, 3 Wolfson Palliative Care Research Centre, Hull York Medical School, University of Hull, Hull, 4 Professor of Palliative Care, Wolfson Palliative Care Research Centre, Hull York Medical School, University of Hull, Hull, 5 Researcher, Wolfson Palliative Care Research Centre, Hull York Medical School, University of Hull, Hull

* annajane01@gmail.com

## Abstract

### Objectives

Unintentional weight loss in older adults is common, with 15–20% of those aged >65 having clinically significant weight loss, associated with increased mortality and morbidity. People with socioeconomic disadvantage are more likely to be overweight but also to become frailer in older age. We explore if socioeconomic status impacts upon patients' experience of unplanned weight loss.

### Methods

Qualitative secondary analysis of 23 semi-structured interviews with older adults from two prior studies i), those at risk of frailty ii) those with cancer. Reflexive thematic analysis was conducted, using the lens of the Nutrition Equity Framework, on anonymised transcripts with formation of themes and subthemes, with relationships between themes investigated.

### Results

Mean age 73 years, range 65–87; 34% male, Index of Multiple Deprivation Quintiles IMD 1 (n=9), IMD 2 (n=4), IMD 3 (n=3), IMD 4 (n=6), IMD 5 (n=1). Three major themes were identified. 1. 'Healthcare Systems'; interactions with either public health or individual healthcare systems influence patient experiences of weight loss. 2. 'Personal Factors'; that influence a patient's view of weight loss and the likelihood of weight loss prompting help-seeking behaviour 3. 'Can I Change?'; patients' perspectives of their ability to implement change. Factors in each of the themes were understood through motivating (reinforcing) and demotivating (balancing) factors.

**Data availability statement:** All relevant data are within the manuscript and its Supporting Information files.

**Funding:** The author(s) received no specific funding for this work.

**Competing interests:** I have read the journal's policy and the authors of this manuscript have the following competing interests: Fliss Murtagh is a National Institute for Health and Care Research (NIHR) Senior Investigator. The views expressed in this article are those of the author(s) and not necessarily those of the NIHR, or the Department of Health and Social Care.

## Conclusions

This study demonstrates that there is structural and individual inequity in individual views, identification, and clinical management of weight loss. The consequences of this disproportionately affect the most deprived, further confounding the inequalities that already exist.

## Introduction

Clinically significant unintentional weight loss in older adults is a component of frailty [1]. Weight loss is common (15–20% ≥65 years), especially in vulnerable groups such as nursing home residents, and is associated with increased mortality, morbidity, and reduced quality of life [2,3]. For community dwelling older adults, fluctuations in weight (both gain and loss) are associated with higher mortality risk compared to stable weight [4]. Weight loss in older, hospitalised adults is associated with adverse outcomes [5] and longer hospital stays [6]. Even in healthy older adults, weight loss has been demonstrated to be associated with an increase in all-cause and cause-specific mortality [7].

In the United Kingdom, malnutrition is a recognised public health problem, costing 15% of the health and social care budget, with older adults accounting for 52% of these total costs [8]. Public health messages, including "thin-ness" as desirable, following high fibre, low-energy-dense diet, and discouragement of snacking are often considered applicable to older adults, and may lead to dietary restrictions [9,10].

However, weight loss in the older adult, without protein supplementation and appropriate graded exercise, can lead to loss of muscle mass in addition to fat mass, potentially accelerating sarcopenia [11]. This is particularly important in the older person, who may disproportionately lose functionality and capacity for independent living [11]. Despite this, healthcare professionals and patients often believe weight loss is a positive or expected part of ageing [12]. Malnutrition is a condition that primary care health professionals feel unsupported and unknowledgeable about managing, with services for obese and overweight adults prioritised over managing malnutrition [12]. Even in high risk populations, nutrition and functional deficits are overlooked, with clinicians misattributing this to ageing, cancer or comorbidities [13].

We also know that socioeconomic status (SES) is associated with weight [14]. More affluent older adults often have a more favourable BMI (body mass index) and weight circumference profile [14], with disadvantaged persons more likely to be overweight in adult life, but also to become frailer in older age [14]. Alongside BMI, higher SES has a positive impact on leisure time physical activity, a strong predictor of a lower future risk of frailty [15]. Poorer SES is associated with accelerated decline in older adults in multiple domains of health, independently of diagnosed health conditions [16]. Within the complex and confounding factors of SES, age and weight loss, qualitative research allows a greater understanding of individual experience. The aim of this study is to explore the impact of SES on older adult's view of their unintentional weight loss.

## Methods

This is a secondary analysis of 23 semi-structured interview transcripts. Interviews were conducted in two studies looking at nutrition and weight loss in older adults. Study 1: Participants were recruited from community-dwelling older adults referred to an integrated care centre run by the community geriatric team. They were eligible if they were identified as having an

electronic Frailty Index score of 0.36 or above. An information sheet describing the study was included in the information pack provided by the service prior to attending the centre. The clinical team identified those who were interested, who were then approached by a designated member of the research team on arrival for their centre appointment, using convenience sampling.

Study 2: Participants were recruited from an inpatient tertiary cancer centre in the Yorkshire and Humber North-East of England. Participants were eligible if they were aged > 70 years with breast, colorectal, lung, prostate, head and neck, or upper gastrointestinal cancers and could give informed consent. Eligible patients were identified and approached by a member of the clinical team, and those who were interested then seen by the researcher. Convenience sampling was used to recruit participants. Interviews were transcribed verbatim and anonymised. Transcripts were analysed as a single dataset.

We used a pragmatic approach, focusing on real-life clinical application and analysed the data following the six phases of reflexive thematic analysis [17]. Familiarisation and initial note taking was conducted by ANC who coded an initial set of 25% of the interviews in a data-driven, inductive approach. These were then independently coded by two members of the research team (AB, MJ). Discrepancies in coding were discussed with the intent of enriching analysis through collaboration, prior to the formation of a codebook. ANC coded the remainder of the dataset, forming both semantic and latent codes using constant comparison. Codes were grouped together to form subthemes and then organised into descriptive themes, each unified by a single, central concept. These candidate themes were discussed and refined (AB, MJ, ANC) to form analytic themes. See online supplementary S1 Table for themes table. Once the themes had been finalised, participant, area-based SES was applied, using the Index of Multiple Deprivation (based on the 2019 Census and postcode [18], Quintile 1 (highest), to 5 (lowest). This allowed for the data to be understood in the light of how participants with different SES related to subthemes, creating a depth and richness in the analysis. SES was applied to the data at this point to minimise any bias that might have been present if this was added before analysing the data. In addition, we applied the lens of the Nutrition Equity Framework to contextualise the participant's lived experience within the wider, structural inequity of nutrition [19].

This framework identifies how social and political structures interact and shape inequity in nutrition, looking at unfairness, exclusion and injustice understood through unequal power dynamics. Understanding how these factors interact within the wider socio-political contexts is important when implementing long-term and sustainable changes. We adapted and simplified this framework to make sense of study results, focusing on the intermediate determinants of nutrition and how this results in individual factors, with an appreciation for the contextual driver of the structural determinants.

Author reflexivity, important to the interaction between researcher and data, was completed throughout the analysis: ANC is university-level educated medical clinician, within three years of qualification. MJ is a senior applied health researcher with 38 years as a medical clinician, mostly within the specialty of palliative care. AB is a clinical academic dietitian.

Both studies were approved by the NHS Ethics Committee (reference 19/LO/1479, reference 18/YH/0470), with studies performed in accordance with ethical standards laid down in the 1964 Declaration of Helsinki and its later amendments, with informed consent, including consent for secondary analysis, obtained from all participants. For this secondary analysis, authors had access to anonymised data that could not identify participants. All participants gave informed written consent which included the use of their anonymised data for future research by authorised researchers.

## Results

Twenty-three semi-structured interview transcripts were analysed. Three themes were generated from the data; 1) Interaction between healthcare systems and the individual, 2) Personal factors impacting on an individual's view of their weight loss and 3) Can I Change?

### Theme 1: Interactions between healthcare systems and the individual

All participants expressed the public health message of healthy eating, implicitly or explicitly appraising the value of their diet against this. Participants living in less deprived areas discussed these more positively, viewing their own diet as in line with the public health messages. This was reflected in a feeling of capability and empowerment in their diet.

> *"Other than… erm… I we've always had skimmed milk, hah, as that's more healthy, but [Dietitians recommend], full fat milk, so will have to look at that, but other than that I seem to be, [Dietitian] seem to be thinking I was doing all the right things really" (Participant 2, Quintile 5)"*

> *"I always have a lot of fruit and veg, and things like that so, it hasn't been hard for me atall" (Participant 3, Quintile 3)*

However, participants living in more deprived areas viewed their own diets negatively, as they did not conform with expected ideals.

> *"But I don't eat salads. Like, I feel awful at work because I take like egg sandwiches or ham sandwiches. They're all sat there with their plastic things full of like salad and different kinds of things. I wished I ate like that like that" (Participant 12, Quintile 2)*

Some participants felt stigmatised by healthcare providers about their weight and felt burdened by the language of these interactions.

> *"You've got to be careful on how you phrase it because there are some that cannot lose [weight] and they're feeling picked on. You know, it's not right"(Participant 21, Quintile 1)*

Conversely, others felt healthcare professionals did not act on their concerns about weight loss, with healthcare professionals felt to minimise nutritional issues leaving participants feeling abandoned.

> *"He just says to me, "you are underweight, but it's better than being overweight." They don't seem bothered about my weight and yet..." (Participant 23, Quintile 2)*

> *"Yeah, that's what I feel like. Like before. I felt a bit abandoned, you know, because it's like there's nothing the matter. So, go home. Then you go home, and you think but why am I still getting these pains in my stomach? You know, why not something probably like the said the IBS [irritable bowel syndrome], but I get really a lot of pain in my back, you know at the bottom of my ribs coming around into my chest like that all the time" (Participant 12, Quintile 2)*

The ability to challenge this attitude and facilitate change was more readily expressed by participants in less deprived areas, who persevered until a better outcome was forthcoming.

> *"The only frustration was this weight loss, lac of energy started to arise, just getting somebody to take it onboard which has now happened, and we have seen the improvement, but the*

*period, I suppose between end of September through to… beginning of this year when I felt a little frustrated because we raised these issues… and we… were… not fobbed-off that's too strong a word, but nothing really materialised" (Participant 1, Quintile 4)*

Where positive interactions with healthcare professionals regarding weight loss were expressed, participants were more often living in less deprived areas, and expressed trust with their healthcare professionals, stating confidence that they would act on their behalf.

*"I think I told them, … [NAME] was my personal nurse, and we poured our heart out to [her] …" (Participant 4, Quintile 4)*

*"No my doctor, my ordinary doctor… Well I told him. I wasn't eating! [laugh] I said I'm kinda losing weight, I'm not eating, ha, yeah, and this is why, this is why he put me for the scan, because, it got to the stage where I wasn't eating solid food" (Participant 5, Quintile 3)*

Interactions with healthcare professionals were described more passively by those in more deprived areas, whereby participants experienced interventions as happening to them, rather than being self-driven:

*"She checked it on the computer. That's why she says I'm putting you... I told you I wasn't keen on them other foods, they're in there on the kitchen, whatever they call them. She said, I'll send you a sample pack and I got those and I got me drinks" (Participant 13, Quintile 1)*

### Theme 2: Personal factors impacting on an individual's view of their weight loss

**Subtheme 1: Rituals and routines.**  Rituals of eating and the traditional 'three meals a day' eating pattern was shared by all participants although language used to describe this differed. This was portrayed as a rigid and inflexible pattern, sometimes reinforced by healthcare professionals.

*"If we're just on our own we have a cup of tea and a biscuit. At lunchtime we have sandwiches and then we have a cooked meal of an evening" (Participant 9, Quintile 4)*

*"But he [Doctor] said, "you must have a regular mealtime" which I do. I'll have a onc..." (Participant 15, Quintile 1)*

This routine was particularly expressed by those in more deprived areas, especially highlighting the importance of Sunday lunch. This tradition was maintained despite challenges, e.g., difficulty cooking or reduced appetite. The ritual of the meal appeared to be almost more important than the eating itself, with strong social implications and family ties.

*Participant: I get breakfast in, when me daughters aren't here. Yeah, I get me Sunday lunch delivered [right] which I pay for but it's very enjoyable" (Participant 17, Quintile 1)*

*"I can cook myself a Sunday dinner and have two mouthfuls and I'll throw it" (Participant 20, Quintile 1)*

*"… I often go [to grandsons] weekend, like a Sunday for me Sunday lunch" (Participant 15,Quintile 1)*

Food was described by all participants as central to socialising, both amongst family and with friends, for celebrations and grief, forming a space in which to maintain these relationships in changing circumstances.

> *"Once a month, they've always... My other sisters have always had... Since my mother died, once a month they'd have lunch" (Participant 22, Quintile 4)*

> *"And we enjoy our food, you know, socialize and things like that" (Participant 8, Quintile 3)*

> *"I mean we have a big family, my husband has quite a big family as well, yes, it has affected us social life, because we can't go out for meals as much as we used to do, erm, not because I can't eat, but because I wasn't feeling very social" (Participant 3, Quintile 3)*

> *"May this year, me and [wife] 44th wedding anniversary and we all went for, for, a family-meal - well, they had a family meal, I just had a beer" (Participant 17, Quintile 1)*

**Subtheme 2: Why I am not worried about my weight loss.** Justifications for not being worried about weight loss were complex, these included being slim throughout adult life, never being particularly interested in food, or not feeling unwell, leading to lack of concern about weight loss in older age.

> *"Participant: I lost a stone… I've always been on the slim side" (Participant 16, Quintile 1)*

> *Not really. No. It was just I'm not a very good eater anyway. You know, so it won't that, but I seemed to age overnight as well" (Participant 9, Quintile 4)*

> *"Oh no, no, no it doesn't play on me mind, no but it could do if you were that type of person" (Participant 5, Quintile 3)*

Effect of stress, grief, and the burden of caring for a loved one, was also associated with weight loss. The link with weight loss did require initial coaxing from the interviewer, and seen as was less concerning, as it was seen as having a causative reason.

> *"Well, I started losing weight because I'd been looking after my husband, who was poorly. I started losing the weight and then he died and I still kept losing the weight. I used to be about 10"10'... I'm seven stone something now and I cannot put it on" (Participant 23, Quintile 2)*

> *"Interviewer: I mean, when did your husband die?...*

> *Participant: Four years ago, August… I think that had something to do with it" ((Participant 16, Quintile 1)*

From those living in more deprived areas, this normalisation of changing food habits was discussed in comparison to others, either other people of the same age or in relation to how family members were as they aged:

> *"Oh just the normal food, but I don't eat a lot. I don't think you do when you get older"(Participant 9, Quintile 4)*

> *"Eating like...I have my dinner, say... I don't have as much dinner as I did, but I think as you get older, everybody is, because you talk to people in here [hospital ward] and they always say they don't eat as much because you're not moving about, you're not at work" (Participant 23, Quintile 2)*

Eating something, even if less than previously, was considered better than eating nothing and helped minimise anxiety. Participants also made positive comparisons to others.

> *"And I see people who're properly poorly like my daughter died of motor neurone disease. And then nine and a half months later my husband died of it as well. So, I have seen stuff and I've seen stuff on the world and when I worked for [private hospital], I've seen stuff there and that is horrendous. And I think just thank your lucky stars, but I still go "Oh my stomach!", you know what I mean and then I think "shut up!"" (Participant 12, Quintile 2)*

> *"We've seen a lot of other people who are a lot more poorly than we are. And the more you come here, there more you listen and think, crikey, I have nothing to worry about, you hear what other have, and there still here and they're still fighting, leading their life as normal as possible" (Participant 3, Quintile 3)*

Some participants were pleased to lose weight, albeit unintentionally. Some were so pleased that they hoped to continue the weight loss intentionally. Whilst others expressed a desire to regain weight.

> *"I'm not bothered about it, not really. It's a good thing to lose it, I'm not saying... that... but I'm not concerned" (Participant 14, Quintile 2)*

> *"I like it, I like being a bit more slimmer. I'm not skinny, skinny, but I could do with putting alittle bit more, but I don't want to put too much on" (Participant 15, Quintile 1)*

> *Husband: "You are about where you should be now in weight terms…*

> *Participant: I have been trying to lose weight for 50 years… I'd like to lose some more weight in the next... this month?" (Participant 22, Quintile 4)*

Participants, particularly living in more deprived areas, having previously been overweight, were used to being actively encouraged to lose weight by healthcare professionals. Being overweight in this context was experienced as a barrier to accessing healthcare services.

> *"So, I said right I'll go on a diet. So, I stuck to the diet every bit I stuck to the diet. He [GP] gave me a sheet, thousand calories a day" (Participant 18, Quintile 1)*

> *Participant: "Well, it's a little different now because I went to see this physiotherapist and she didn't even weigh me or measure me. She said, "How tall are you?" I said, "5ft 4." "How much do you weigh?" I told her, I said, "15 2." She said, "Oh, your BMI is 36 you've got to have it under, you've got to lose weight.*

> *Interviewer: For the surgery?*

> *Participant: No, just to see the specialist" (Participant 20, Quintile 1)*

For all participants, there was a resignation to a loss of appetite and lower level of health associated with ageing.

> *"Resigned? You can't help but be resigned, I mean, I can't do the things that I used to donow, I'm reconciled to it" (Participant 8, Quintile 3)*

> *"I mean, I get tired but I think that's old age. I don't think that's anything to do with myhealth. As I say, I've nothing to complain about" (Participant 23, Quintile 2)*

**Subtheme 3: Why I am worried about my weight loss.** In general, concerns arose only with a very noticeable or rapid change in appearance or feeling ill. Rapid weight loss was viewed as concerning amongst all participants, stimulating help-seeking behaviour and a reason to seek medical assistance.

*"No it was the weight loss love, yeah. I didn't, I didn't realise how much I was losing, andhow quickly… it didn't, it didn't strike me at first, and then I realised, then I thought you know this is too much, I've got to do something an I went to the doctors…" (Participant 6, Quintile 1)*

*"No, I thought there was something wrong. … she's losing weight too fast and that to me was wrong… half of her clothes didn't fit her" (Participant 10 and Carer of Participant 9, Quintile 4)*

*"Started at nine stone three, and, and then this seven started appearing, and I'm thinkingthis really isn't good news, you know…" (Participant 1, Quintile 4)*

*"Well, only if it did drop very drastically, then I'd know there's something wrong. It's comingoff bit by bit, so I know… [it's nothing to worry about]… but if I get up tomorrow and I'mabout five stone then I'd think, "Oh my God, what's wrong?"" (Participant 21, Quintile 1)*

Face to face consultations were preferred over telephone consultations, allowing for physical cues to action to be seen more easily.

*"You see the problem is you can't go in to see your GP [General Practitioner] to really explain it or show him how you're losing weight, so I suppose he's at a disadvantage, so there's everything against it really, as far as that bit goes" (Participant 5, Quintile 3)*

*When he [doctor] lifted my top, he could see how much weight I'd lost, and my clothes was hanging down. Anyway, he went "oh goodness gracious!". I mean, he felt down the side of my stomach and it hurt, and he went. "Oh, I'm sending you to a hospital for x-rays" (Participant 12, Quintile 2)*

In addition to weight loss, visible reduction in oral intake was another cause for concern, although mainly for participants in higher deciles.

*"…that I could no longer eat the volume of food… that I was eating… I sort of when down formy dinner to a salad plate, and then to a tea plate…" (Participant 1, Quintile 4)*

**Subtheme 4: Weight loss and the importance of appetite.** Participants' perception of appetite and its relationship to their diet and weight varied. Some participants expressed an understanding of the importance of appetite in weight loss:

*"But I was really pleased, it was only a bit recently, when I was able to get suddenlystarted to get my appetite back" (Participant 2, Quintile 5)*

Although some participants in more deprived areas expressed difficulty in understanding why they were losing weight when there was sufficient food available.

***Husband interjects:*** *"She has cupboard full of food and she won't touch it" (Participant 18,Quintile 1)*

*"I don't know. I just cook it and just love nothing. I don't want this" (Participant 16, Quintile1)*

Going to a restaurant for a meal was seen to have a positive effect on appetite, however for participants in more deprived areas there was a fear of food waste, which impacted on this effect:

*"Yeah, I would eat it, but I would leave some of it Yeah. You know, when I go for a meal, Iput on what I think that I can eat" (Participant 15, Quintile 1)*

## Theme 3: Can I change?

**Subtheme 1: Patient perceptions of barriers to change.** A lack of flexibility in dietary patterns restricted participants ability to make changes in their eating. A determination to maintain normality despite needing to make changes was seen, with a feeling of frustration at lack of knowledge on how to gain weight, and perseverance of large portion sizes despite finding them off putting.

*"No, not really. I was losing weight and I couldn't put weight [on], and I still can't put weighton. No matter what I eat, I cannot put weight on. I don't worry about it now. I won't say I'ma worrier, I'm not, but I just think about it - "I wish I could put some weight on" (Participant23, Quintile 2)*

*"No not really, just keep as normal as you had before" (Participant 3, Quintile 3)*

Participants living in more deprived areas particularly expressed difficulty with portion sizes, this was reinforced by negative comments regarding wasting food from relatives.

*"Oh no, it's just the way I eat has changed and portions have changed. I don't... I can't eat it if I have a plate loaded with food. I can't eat that"… "If I put too much on, he goes, "Your eyes are bigger than your belly" sort of thing" (Participant 15, Quintile 1)*

*"I just can't get it down me. I feel sick, not sick but I said, "I'm full." (Participant 13, Quintile 1)*

This lack of flexibility also placed the emphasis on ensuring that participants had at least one, hot, square meal a day. This was seen as more important that the amount consumed in its entirety.

*"But, we always make sure that my mam has a full meal one day. There's always food there,whatever. We'll go out. I go out on a Saturday with my mam, basically every week-end forthe last two or three years, we always have a full meal and whatever" (Son of Participant13, Quintile 1)*

A sense of disengagement was felt with weight changes as participants aged, as it was felt to be predetermined by the patterns set by their family members before them.

*"Yes. I've always had weight problems because my mum's family they're all big. All my mam's sisters they've all been big. My mam was big, my grandma was really big. My cousin, she's big, but she's real tall. So, I just think that's in the genes. I've always had problems with my*

*weight. I was bullied at school with my weight and things like that. I have tried" (Participant 20, Quintile 1)*

There was a sense, particularly for those in more deprived areas, of food guilt. This was founded in participants trying and failing to conform to the socially normal eating patterns. This was also felt in guilt of being unable to eat as 'healthily' as others around them, attributed to food poverty as a child.

*"I don't eat breakfast, I don't eat lunch, and sometimes for my tea I'll just have some toast ora sandwich. I know it's wrong" (Participant 20, Quintile 1)*

*"But you see I was the oldest of nine. We never had any food at home. [Yeah] And I think it stems from that, you know" (Participant 12, Quintile 2)*

Those in more deprived areas also expressed a belief that they deserved ill health in older age due to a lifetime of not eating as they should:

*"I did. And when [Doctor] came to see me, he looked at me and when "Huh" and I went "I know!"and I said it's a lifetime of a bad diet. It's my own fault" (Participant 12, Quintile 2)*

Whilst some shared a feeling of disillusionment with public health messages, finding them inaccessible and off-putting:

*"They're saying you shouldn't eat meat. You shouldn't eat this. Might just as well give up"(Participant 21, Quintile 1)*

Some of the challenges of eating in older age were affected by loss of independence. This was mainly expressed by those living in more deprived areas. Independence involved both being able to feed oneself and the financial independence to be able to afford to cook and dine out.

*Yes. Now and again, we've been going out for meal and I said to him, we'll have to stop tha-tit's too expensive" (Participant 15, Quintile 1)*

*"I use to make my own [meals] but it's not worth it for the money" (Participant 13, Quintile1)*

Independence was also affected by confidence. Lack of confidence impacted on activities that were once taken for granted and now required help from others.

*"No, well, I maybe warm some soup up. The thing is, it's only this last month or more I'm abit dithery. And [son] don't, [son] would simply come and sort dinner" (Participant 13,Quintile 1)*

Both physical and mental health impacted on participants' ability to eat. This ranged from physical health conditions affecting ability to chew, to changes in cognition affecting participants' ability to remember their own weight loss.

*"And me teeth...? I couldn't keep them in, with false teeth which just fell out because I've lost-that much weight. Round my mouth and my face, me teeth were too big for me mouth"(Participant 15, Quintile 1)*

*"Yeah, we mentioned it on several...I mentioned it on several occasions so I know I mean,-yeah, I know that [patient]'s memory is not so good so he can't remember and I now I'vehave mentioned it…" (Friend and carer of Participant 11, Quintile 2)*

Experiences of ill-health had repercussions for psychological well-being surrounding eating. This was expressed with a loss of spontaneity and freedom with eating and the effect on appetite and motivation to eat.

*"Yes I am a bit more careful of what I am eating, and where I eat, you know, I know what I've cooked myself, and how I've cooked it, whereas as before you don't really think of it do you? And things like fruit and that I'm a bit more careful about washing them, whereas before I've just grabbed an apple and eaten it" (Participant 3, Quintile 3)*

*"Appetite, I had no appetite. I just didn't think I could manage it and then we all came back here with the grandchildren and had a letter from [hospital] and "the surgery was successful, you're now cancer-free". I just cried [I bet] and I started eating again. I've got my appetite now" (Participant 17, Quintile 1)*

**Subtheme 2: Perceptions and experiences of being able to implement change.** Support of family members and informal carers was instrumental in identifying, implementing, and advocating for change.

*"When other people started to mentioning it, how thin I was getting, that opened my eyes a little bit, you know, erm" (Participant 6, Quintile 2)*

*"Well, I would have taken her back to the doctors. [the GP?]. Yeah" (Participant 10, carer of Participant 9, Quintile 4)*

The caring actions of family members often revolved around cooking for participants, adding to the burden of informal caring responsibilities. Some participants living in more deprived areas, utilised the community around them, paying for the delivery of food, and viewed this as cheaper than cooking for themselves. This also enabled participants to feel in control of their food choices, with family preparing preferred foods, with food freedom seen as important.

*Interviewer: "Your main meal is from them [the neighbour].*

*Participant: Yeah, and I pay her for that. Which I don't mind because it costs me more if I put the oven on." (Participant 15, Quintile 1)*

*"She's on 12-hour days on tomorrow and she's going to come and take me shopping on Wednesday with me wheelchair" (Participant 17, Quintile 1)*

An enjoyment of the physical changes associated with weight gain were expressed by some participants in more deprived areas, reinforced by their family members.

*"And the girls see the difference. Last time I went out, cos I go to [place name], they said "Oh you look wonderful, [name]". Cos last time they saw me, I was like death warmed up basically. Eyes were sunken, cheeks were in here. So, that's coming back now" (Participant 17, Quintile 1)*

Participants living in less deprived areas shared a belief in patient-centred healthcare, whereby they could initiate and ask for guidance from their healthcare professionals and vocalised the importance of education to facilitate engagement and pro-activeness in health.

*"I would have wanted them to really refer her to a hospital consultant" (Participant 10, carer of Participant 9, Quintile 4)*

*"I think it, it actually does you a lot of good to actually vocalise and make you sort of thinkthrough it, we've both been in education both our lives, and I was quite interested from thatpoint of view" (Participant 1, Quintile 4)*

Some participants were flexible in their diets, either driving change through self-help measures or accepting that the goalposts of nutrition were different in the context of weight loss.

*"He don't eat a lot of...most of your diet is meat and fried stuff. But at least he's eating"(-Friend and carer of Participant 11, Quintile 2)*

Future goals were important in motivating participants to change their health and diet. These included luxuries such as travelling abroad, functionality such as maintaining muscle mass and access to further treatments in their cancer pathway. These were individual and varied but allowed for a goalpost for participants to work towards.

*"Yes. I think, as well, apart from me getting better and feeling better we'd like another three-years of long holidays, wouldn't we?" (Participant 22, Quintile 4)*

*"Oh, no, I don't want to lose me muscle" (Participant 15, Quintile 1)*

*"But make sure you eat as nutritionally as you can, which will help you feel better, an cope-with the treatment better" (Participant 3, Quintile 3)*

## Application of the nutrition equity framework

Our application of the Nutrition Equity Framework [19] is summarised schematically in Fig 1. The structural societal inequity in the way people eat influences their attitudes and behaviours around unintentional weight loss in older age. The consequences disproportionately affect those that are more deprived, further confounding already existing inequities. This perpetuates further with those from more deprived areas being less likely to engage in health-seeking behaviours and more passive in consultations with healthcare professionals. This is confounded by the knowledge that doctors have been shown to give more information and more emotional support in a shared decision-making style with patients with a higher SES [20].

Our research shows collusion between healthcare professionals and participants with the view that all weight loss is positive, which is more pronounced among participants from more deprived areas and those who had been overweight in adult life. These factors result in a vicious cycle of increased unfairness, injustice, and exclusion of those from more deprived areas.

## Discussion

To our knowledge, this is the first qualitative study to stratify older adult's experience of nutrition and weight loss by socioeconomic status, allowing us to further understand the barriers in clinical management of weight loss in this population. This study demonstrates structural societal inequity in the way people approach and experience food and diet, which impacts on their attitudes and behaviours around unintentional weight loss.

Our findings are consistent with previous research identifying the messages from adulthood, including concerns about weight gain, are carried forward into old age [9,10,21]. Older adults from lower socioeconomic backgrounds were more likely to feel stigma with regards to their diet and assign self-blame for choices that deviate from public health messages. Despite

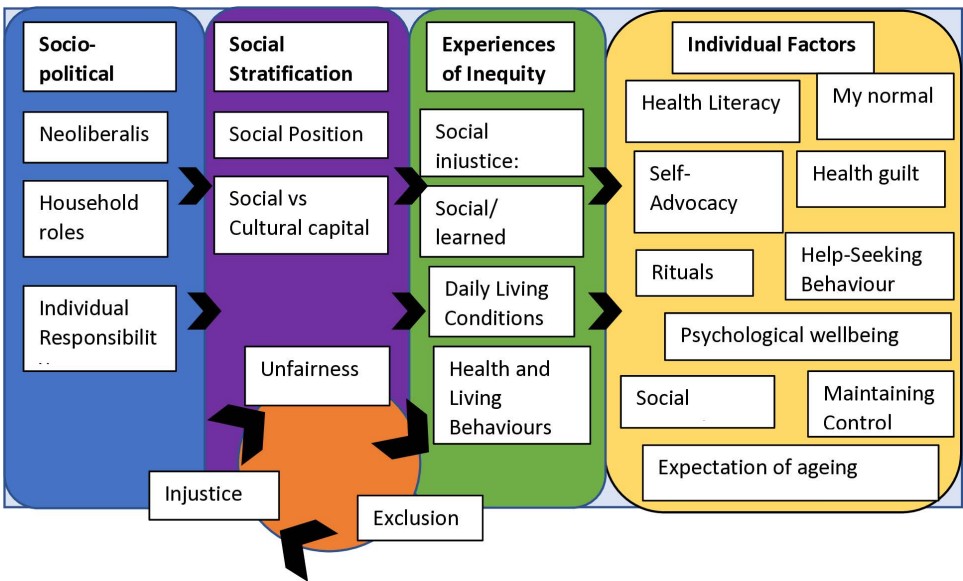

**Fig 1. Adaptation of the Nutrition Equity Framework in the context of unintentional weight loss** [ 40].

the good intentions of health policy, this can lead to feelings of shame and inferiority, particularly in those from lower socioeconomic decile [22–26]. Previous research into weight gain has demonstrated that in older adults, the prevalence of frail obesity was higher in those from lower SES and increased in prevalence by 1.49 for each additional social disadvantage [27]. Having a higher social status has been shown to buffer the psychological impact of weight stigma [24,28]. Carrying this forward into older age, those from lower socioeconomic deciles carried stigma and guilt into their later life, mitigating concern they had for unintentional weight loss.

Participants from more deprived areas were less able to challenge healthcare professionals and less likely to alter the outcome of the consultation with their concerns. Passivity in healthcare consultations has been demonstrated in different contexts in those from lower socioeconomic backgrounds, whilst physician bias can affect recognition of healthcare needs [29–31]. Our findings in weight loss adds to existing evidence, highlighting that tailoring information giving and consultation style is important to address these differences [31–33]. Adding to this, our study highlights the importance of not colluding with older adults, particularly those from higher socioeconomic deciles, that all weight loss is positive [29].

Experiences of unintentional weight loss was heavily influenced by external factors, with a shared feeling of resignation amongst participants to lower quality of health as you age [34,35]. This shared feeling of inevitability was a barrier to recognising concern in weight loss. Promoting a positive perception of ageing has been shown to enhance older adult's quality of life [36] and could be utilised to encourage positive weight change.

For participants in lower SES, rituals of eating, including the implications for socialising and the feeling of normality was a large factor in maintaining rigidity in dietary practice, impacting on the ability to gain weight. Confounding this was a reluctance to alter portion size or waste food. This has interesting parallels in barriers to studies in weight loss in lower SES [37,38]. Weight loss in isolation was not considered concerning [10] only triggering health seeking behaviour when it was associated with other factors of ill health [39].

## Clinical and research implications

Unintentional weight loss is sometimes a clinically important indication of serious illness which is often overlooked by the person and their clinician alike. For older adults with unintentional weight loss, time should be taken to understand their current diet. This involves understanding what they mean by a "good diet", rituals they observe, and changes they feel capable of making. We may need to remove the guilt of eating higher calorie foods and snacking, releasing them from the messages that have been reinforced since childhood. Older people must be given permission to eat as it suits them, removing the rigidity of set meals time and offer solutions, such as using smaller plates to make portions look less off-putting.

Particularly for those living in more deprived areas, we must not stigmatise how they eat, and take time to undo the negative comparisons that they make to others. For all, but particularly for those from higher deciles, we must take care not to collude on weight loss, ensuring an understanding of the differences between intentional and unintentional weight loss and that patient views on weight loss are appropriate.

Patients, especially those living in more deprived areas, may not initiate conversations surrounding weight loss, believing that healthcare providers would mention it if concerned. Weight loss should be considered as a symptom that needs treating in the context of advanced disease or – contrastingly - after negative investigations. We must not assume that weight loss will be self-managed by patients. Giving permission to alter diets can help to introduce flexibility, empowering patients to make sustainable changes in their diet.

Future research should aim to understand the most effective and sustainable changes in diet for older adults and to further understand how language surrounding weight loss and nutrition impacts on individual outcomes.

## Strengths and limitations

This study provides detailed, in-depth discussion of patient experience surrounding their weight loss. It is one of the few studies to consider the context of socio-economic status in relation to these experiences. Rich data was included in the study, with a diversity of participants (age, sex, SES), generating a wide range of views.

Limitations include the assignment of SES to participants by postcode. This means that SES is by an area-based measure as opposed to an individual measure and may not be entirely representative of individual status. The interviews were initially conducted to answer a different research question; therefore, all the pertinent information may not have been provided during data collection. Data were only collected on a small subset of the population in one geographic area; therefore, the findings may not be universally transferable to the wider population although we anticipate these findings will be applicable beyond our study settings. Additionally, the study was limited by the heterogeneity of the participants as White British.

## Conclusion

The importance of socioeconomic factors in health is increasingly apparent. This study adds to current research on patients' perspectives with regards to weight loss and nutrition. Socioeconomic status affects patients' views of their own weight loss, how they engage with healthcare professionals and the outcome of these patients bringing forward their concerns. Healthcare providers should consider SES when tackling nutrition, particularly focusing on addressing underlying health beliefs and capacity to change. Care must be taken to undo unhelpful health messages and redefine the goalposts for weight and nutrition in older adults.

### Key messages

**What is already known on this topic.**  It is already known that weight loss has an association with morbidity and mortality in older adults. It is also known that socioeconomic status influences weight throughout life.

**What this study adds.**  This study adds evidence of the impact of socioeconomic status on older adult's views of unintentional weight loss. This is explored through differing perceptions of diet, public health messages and interactions with healthcare professionals. Older adults from more deprived areas are more likely to feel stigma surrounding their diet, less likely to be flexible in their dietary patterns and less likely to advocate for themselves in interactions with healthcare professionals.

**How this study might affect research, practice or policy.**  This study advocates for more research on how best to change eating practices in older age to combat weight loss and accelerated sarcopenia. It provides evidence for the negative impact of public health messages on weight loss in older adults and suggests that to positively change weight loss in older adults, we must remove food guilt and tailor dietary advice to an individual's circumstances.

## Supporting information

**S1 Table.  Table of themes and subthemes used during methodology.**
(ZIP)

**S1 Fig.  The Nutrition Equity Framework [40]. This is figure adapted for use during the data analysis40. The Nutrition Equity Framework: Nisbett N, Harris J, Backholer K, Baker P, Jernigan VBB, Friel S. Holding no-one back: The Nutrition Equity Framework in theory and practice. Global Food Security. 2022;32:100605.**
(TIF)

## Contributorship statement

The project was designed by AB and MJ. Data collection was conducted by AB and UN. Data analysis was conducted by ANC. ANC wrote the manuscript. All authors revised the manuscript critically. ANC, AB and MJ have overall responsibility for the final content.

## Author contributions

**Conceptualization:** Anna Newton-Clarke, Alex F Bullock, Miriam J Johnson, Fliss EM Murtagh.

**Data curation:** Alex F Bullock, Miriam J Johnson, Ugochinyere Nwulu.

**Formal analysis:** Anna Newton-Clarke, Alex F Bullock, Miriam J Johnson.

**Methodology:** Anna Newton-Clarke, Alex F Bullock, Miriam J Johnson.

**Project administration:** Alex F Bullock.

**Writing – original draft:** Anna Newton-Clarke.

**Writing – review & editing:** Alex F Bullock, Miriam J Johnson, Ugochinyere Nwulu, Fliss EM Murtagh.

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
