## [Decision Letter · Decision Letter 0]

2 Apr 2024

PONE-D-23-16906Socioeconomic status and older adult’s experiences of weight loss: a qualitative secondary analysisPLOS ONE

Dear Dr. Newton-Clarke,

Thank you for submitting your manuscript to PLOS ONE. After careful consideration, we feel that it has merit but does not fully meet PLOS ONE’s publication criteria as it currently stands. Therefore, we invite you to submit a revised version of the manuscript that addresses the points raised during the review process.

Overall, I agree with the comments of Reviewer 1--please attend carefully to their comments both in the review itself and in the annotated manuscript. Additionally, please better define the inclusion criteria (namely, please include how "at risk for frailty" was operationally defined) and include more information on the demographic characteristics of the participants (e.g., gender, race/ethnicity) to better contextualize the findings. Please include information on how participants were recruited and the purpose of the primary study for which they were originally recruited. Finally, please carefully proofread the manuscript for spelling and grammar.

We look forward to receiving your revised manuscript.

Kind regards,

Emily Lund

Academic Editor

PLOS ONE

[I have read the journal's policy and the authors of this manuscript have the following competing interests: Fliss Murtagh is a National Institute for Health and Care Research (NIHR) Senior Investigator. The 

views expressed in this article are those of the author(s) and not necessarily those of the NIHR, or 

the Department of Health and Social Care.]. 

Additional Editor Comments:

Overall, I agree with the comments of Reviewer 1--please attend carefully to their comments both in the review itself and in the annotated manuscript. Additionally, please better define the inclusion criteria (namely, please include how "at risk for frailty" was operationally defined) and include more information on the demographic characteristics of the participants (e.g., gender, race/ethnicity) to better contextualize the findings. Please include information on how participants were recruited and the purpose of the primary study for which they were originally recruited. Finally, please carefully proofread the manuscript for spelling and grammar.

Reviewers' comments:

Reviewer's Responses to Questions

**Comments to the Author**

1. Is the manuscript technically sound, and do the data support the conclusions?

Reviewer #1: Yes

2. Has the statistical analysis been performed appropriately and rigorously? 

Reviewer #1: Yes

3. Have the authors made all data underlying the findings in their manuscript fully available?

Reviewer #1: Yes

4. Is the manuscript presented in an intelligible fashion and written in standard English?

Reviewer #1: Yes

5. Review Comments to the Author

Reviewer #1: Reviewer comments PONE-D-23-16906

Introduction

The authors have provided a very limited review of the literature with although the argument for the social and scientific value is logical. This can be improved on by correcting minor grammatical errors and ensuring the quality of literature referenced is more recent data. The statement for the aim should be revised as it contains a repetition that in my view is unnecessary.

Methods

This was a secondary analysis of anonymized data from previously concluded research.

The authors have provided a very thorough description of the process used in data analysis. The diversity among authors would offer adequate diversity The engagement between the authors seems quite robust and is adequate.

Results

The authors have presented results adequately with a good number of participant statements to illustrate the themes and sub themes.

Discussion

This could be improved with comparison with more studies, although few studies have looked directly at the subject, there is a lot of research in behaviour change and diet in weight gain. However, the authors have a clear section describing the implications of their findings for clinical practice and further research. They have also clearly outlined the limitations of this study in looking at the subject especially as it concerns exhaustive exploration of the subject for this study. This fact may have limited the findings but is an indication for further research.

Conclusion

This clearly outlines the main findings and outlines the need for behaviour change in the attitude of clinicians in consultation with older adults with unintentional weight loss.

Clinicians are obliged to empower patients further in correcting misinformation and misconceptions about weight loss.

References

As mentioned earlier, the authors should present more recent data, and consider including more studies in the presentation of the scientific and social value for this study. The comparison with other research can be improved upon.

The style of referencing is correctly done

6. PLOS authors have the option to publish the peer review history of their article (what does this mean? ). If published, this will include your full peer review and any attached files.

**Do you want your identity to be public for this peer review?** For information about this choice, including consent withdrawal, please see our Privacy Policy .

Reviewer #1: No

---

## [Author Response · Author response to Decision Letter 0]

26 Jun 2024

Please ensure that your manuscript meets PLOS ONE's style requirements, including those for file naming. The manuscript has been formatted as per requirements

Please include how "at risk for frailty" was operationally defined Extra information included in to the methods section- “ They were eligible if they were identified as having an electronic Frailty Index score of 0.36 or above”

Include more information on the demographic characteristics of the participants (e.g., gender, race/ethnicity) to better contextualise the findings. Addition to the methods of more demographic data:

Study 1: Average age was 77.4 years, 14% were male. All participants were white British.

Study 2: Median age of participants was 75 years, 75% male. All participants were white British.

Please include information on how participants were recruited and the purpose of the primary study for which they were originally recruited Additional information added to the methods on the sampling approach for original interview participants:

Study 1: Participants were recruited from community-dwelling older adults referred to an integrated care centre run by the community geriatric team. They were eligible if they were identified as having an electronic Frailty Index score of 0.36 or above. An information sheet describing the study was included in the information pack provided by the service prior to attending the centre. The clinical team identified those who were interested, who were then approached by a designated member of the research team on arrival for their centre appointment, using convenience sampling.

Study 2: Participants were recruited from an inpatient tertiary cancer centre in the Yorkshire and Humber North-East of England. Participants were eligible if they were aged > 70 years with breast, colorectal, lung, prostate, head and neck, or upper gastrointestinal cancers and could give informed consent. Eligible patients were identified and approached by a member of the clinical team, and those who were interested then seen by the researcher. Convenience sampling was used to recruit participants.

Please carefully proofread the manuscript for spelling and grammar The manuscript has been carefully reread as per requirements

Please include captions for your Supporting Information files at the end of your manuscript, and update any in-text citations to match accordingly. Please see our Supporting Information guidelines for more information: http://journals.plos.org/plosone/s/supporting-information. Captions have been added for supporting information files

The statement for the aim should be revised as it contains a repetition that in my view is unnecessary. Statement updated and revised to be clearer and more concise:

The aim of this study is to explore the impact of SES on older adult’s view of their unintentional weight loss

Include more recent data on

the public health expenditure

on malnutrition Updated with more recent data:

“ In the United Kingdom, England, malnutrition is a recognised public health problem, costing 15% of the health and social care budget with older adults accounting for 52% of these total costs. (8)”

Reference: 8. Stratton R. Managing Malnutrition to improve lives and save money 2018.

The authors have provided a very limited review of the literature with although the argument for the social and scientific value is logical. This can be improved on by correcting minor grammatical errors and ensuring the quality of literature referenced is more recent data

As mentioned earlier, the authors should present more recent data, and consider including more studies in the presentation of the scientific and social value for this study. The comparison with other research can be improved upon. Literature review expanded and more up to date references included with more references included in the discussion :

4. Alharbi TA, Paudel S, Gasevic D, Ryan J, Freak-Poli R, Owen AJ. The association of weight change and all-cause mortality in older adults: a systematic review and meta-analysis. Age and Ageing. 2020;50(3):697-704.

5. Sullivan DH, Bopp MM, Roberson PK. Protein-energy undernutrition and life-threatening complications among the hospitalized elderly. J Gen Intern Med. 2002;17(12):923-32.

6. Neumann SA, Miller MD, Daniels L, Crotty M. Nutritional status and clinical outcomes of older patients in rehabilitation. J Hum Nutr Diet. 2005;18(2):129-36.

7. Hussain SM, Newman AB, Beilin LJ, Tonkin AM, Woods RL, Neumann JT, et al. Associations of Change in Body Size With All-Cause and Cause-Specific Mortality Among Healthy Older Adults. JAMA Network Open. 2023;6(4):e237482.

8. Stratton R. Managing Malnutrition to improve lives and save money 2018.

13. Bullock AF, Patterson MJ, Paton LW, Currow DC, Johnson MJ. Malnutrition, sarcopenia and cachexia: exploring prevalence, overlap, and perceptions in older adults with cancer. European Journal of Clinical Nutrition. 2024.

15. Kheifets M, Goshen A, Goldbourt U, Witberg G, Eisen A, Kornowski R, et al. Association of socioeconomic status measures with physical activity and subsequent frailty in older adults. BMC Geriatrics. 2022;22(1):439.

21. Jackson SE, Holter L, Beeken RJ. ‘Just because I’m old it doesn’t mean I have to be fat’: a qualitative study exploring older adults’ views and experiences of weight management. BMJ Open. 2019;9(2):e025680.

22. Simons AMW, Houkes I, Koster A, Groffen DAI, Bosma H. The silent burden of stigmatisation: a qualitative study among Dutch people with a low socioeconomic position. BMC Public Health. 2018;18(1).

30. Turner AJ, Francetic I, Watkinson R, Gillibrand S, Sutton M. Socioeconomic inequality in access to timely and appropriate care in emergency departments. Journal of Health Economics. 2022;85:102668.

31. Arpey NC, Gaglioti AH, Rosenbaum ME. How Socioeconomic Status Affects Patient Perceptions of Health Care: A Qualitative Study. Journal of Primary Care & Community Health. 2017;8(3):169-75.

32. Coupe N, Cotterill S, Peters S. Tailoring lifestyle interventions to low socio-economic populations: a qualitative study. BMC Public Health. 2018;18(1):967.

33. Eggink E, Hafdi M, Hoevenaar-Blom MP, Richard E, Charante EPMv. Attitudes and views on healthy lifestyle interventions for the prevention of dementia and cardiovascular disease among older people with low socioeconomic status: a qualitative study in the Netherlands. BMJ Open. 2022;12(2):e055984.

34. Sargent-Cox K. Ageism: we are our own worst enemy. International Psychogeriatrics. 2017;29(1):1-8.

35. Fernández-Jiménez C, Dumitrache CG, Rubio L, Ruiz-Montero PJ. Self-perceptions of ageing and perceived health status: the mediating role of cognitive functioning and physical activity. Ageing and Society. 2024;44(3):622-41.

36. Velaithan V, Tan M-M, Yu T-F, Liem A, Teh P-L, Su TT. The Association of Self-Perception of Aging and Quality of Life in Older Adults: A Systematic Review. The Gerontologist. 2023;64(4).

37. McCoy CA, Johnston E, Hogan C. The impact of socioeconomic status on health practices via health lifestyles: Results of qualitative interviews with Americans from diverse socioeconomic backgrounds. Social Science & Medicine. 2024;344:116618.

38. Siu J, Giskes K, Turrell G. Socio-economic differences in weight-control behaviours and barriers to weight control. Public Health Nutrition. 2011;14(10):1768-78.

Please include your updated Competing Interests statement in your cover letter; we will change the online submission form on your behalf. This has been included in the cover letter and updated on the manuscript to reflect that this does not alter adherence

This could be improved with comparison with more studies, although few studies have looked directly at the subject, there is a lot of research in behaviour change and diet in weight gain. Thank you, we have added a paragraph in discussion with more recent references and brief discussion of the literature on weight gain

Our findings are consistent with previous research identifying the messages from adulthood, including concerns about weight gain, are carried forward into old age [9, 10, 21]. Older adults from lower socioeconomic backgrounds were more likely to feel stigma with regards to their diet and assign self-blame for choices that deviate from public health messages. Despite the good intentions of health policy, this can lead to feelings of shame and inferiority, particularly in those from lower socioeconomic decile [22-26] Previous research into weight gain has demonstrated that in older adults, the prevalence of frail obesity was higher in those from lower SES and increased in prevalence by 1.49 for each additional social disadvantage.[27] Having a higher social status has been shown to buffer the psychological impact of weight stigma. [28] [24] Carrying this forward into older age, those from lower socioeconomic deciles carried stigma and guilt into their later life, mitigating concern they had for unintentional weight loss.

---

## [Decision Letter · Decision Letter 1]

8 Dec 2024

PONE-D-23-16906R1Socioeconomic status and older adult’s experiences of weight loss: a qualitative secondary analysisPLOS ONE

Dear Dr. Newton-Clarke,

Thank you for submitting your manuscript to PLOS ONE. After careful consideration, we feel that it has merit but does not fully meet PLOS ONE’s publication criteria as it currently stands. Therefore, we invite you to submit a revised version of the manuscript that addresses the points raised during the review process.

The work is interesting and potentially appropriate for publication in PLOS ONE. However, reviewers #2 and #4 have raised a number of concerns that prevent us from accepting the manuscript in its current form. We would like to be able to reconsider the manuscript and hope you can successfully address the concerns outlined below by the reviewers.

Key Areas for Improvement:

Kindly strengthen the background of your paper (Reviewer #4).Address the trustworthiness issue that Reviewer #4 brought up in relation to the methods section.Review carefully for grammatical and punctuation errors, especially following citations (Reviewer #2).

In summary, I encourage you to address all the reviewers' comments and make the necessary revisions, particularly in improving the abstract, introduction, method, and discussion sections. I look forward to reviewing your revised manuscript.

We look forward to receiving your revised manuscript.

Kind regards,

Godwin Banafo Akrong, Ph.D.

Academic Editor

PLOS ONE

Journal Requirements:

Reviewers' comments:

Reviewer's Responses to Questions

**Comments to the Author**

1. If the authors have adequately addressed your comments raised in a previous round of review and you feel that this manuscript is now acceptable for publication, you may indicate that here to bypass the “Comments to the Author” section, enter your conflict of interest statement in the “Confidential to Editor” section, and submit your "Accept" recommendation.

Reviewer #2: (No Response)

Reviewer #3: All comments have been addressed

Reviewer #4: All comments have been addressed

2. Is the manuscript technically sound, and do the data support the conclusions?

Reviewer #2: Yes

Reviewer #3: Yes

Reviewer #4: Partly

3. Has the statistical analysis been performed appropriately and rigorously? 

Reviewer #2: Yes

Reviewer #3: Yes

Reviewer #4: Yes

4. Have the authors made all data underlying the findings in their manuscript fully available?

Reviewer #2: Yes

Reviewer #3: Yes

Reviewer #4: Yes

5. Is the manuscript presented in an intelligible fashion and written in standard English?

Reviewer #2: Yes

Reviewer #3: Yes

Reviewer #4: Yes

6. Review Comments to the Author

Reviewer #2: The study acknowledges differences in participants’ experiences based on socioeconomic status (SES), such as those in more deprived areas feeling more passive in healthcare interactions or experiencing greater food-related guilt. However, it does not adequately explore the reasons behind these differences or their connections to structural inequities. A more thorough examination of these underlying causes and their implications would enhance the study’s contribution to understanding SES-related disparities.

Additionally, the manuscript would benefit from a careful review for grammatical and punctuation errors, especially following citations.

Reviewer #3: 1. The manuscript is technically sound, and the data do support the conclusions.

2. The statistical analysis has been performed appropriately and rigorously.

3. The authors have made all data underlying the findings in their manuscript fully available.

4. The manuscript is presented in an intelligible fashion and written in standard English.

Reviewer #4: Review Comments

Abstract: Lacks clarity

Background: Not strong

Methods:Not explicit mainly the trustworthiness of the study is not well stated.

Results;Lacks logical flow

Discussion: Not comprehensive and lacks theoretical and practical implications

7. PLOS authors have the option to publish the peer review history of their article (what does this mean? ). If published, this will include your full peer review and any attached files.

**Do you want your identity to be public for this peer review?** For information about this choice, including consent withdrawal, please see our Privacy Policy .

Reviewer #2: **Yes: ** Anastasia Akosua Asantewaa

Reviewer #3: **Yes: ** Kassa Demissie Abdi (PhD)

Reviewer #4: No

---

## [Author Response · Author response to Decision Letter 1]

23 Feb 2025

We thank the reviewer for their comments, and are grateful for the time they have spent reviewing

our paper. We believe that their comments have improved the quality of our manuscript, and we

look forward to addressing any further feedback. Thank you.

---

## [Decision Letter · Decision Letter 2]

5 Mar 2025

Socioeconomic status and older adult’s experiences of weight loss: a qualitative secondary analysis

PONE-D-23-16906R2

Dear Dr. Newton-Clarke,

We’re pleased to inform you that your manuscript has been judged scientifically suitable for publication and will be formally accepted for publication once it meets all outstanding technical requirements.

Kind regards,

Godwin Banafo Akrong, Ph.D.

Academic Editor

PLOS ONE

Additional Editor Comments (optional):

Reviewers' comments:

Reviewer's Responses to Questions

**Comments to the Author**

1. If the authors have adequately addressed your comments raised in a previous round of review and you feel that this manuscript is now acceptable for publication, you may indicate that here to bypass the “Comments to the Author” section, enter your conflict of interest statement in the “Confidential to Editor” section, and submit your "Accept" recommendation.

Reviewer #2: All comments have been addressed

Reviewer #3: All comments have been addressed

2. Is the manuscript technically sound, and do the data support the conclusions?

Reviewer #2: (No Response)

Reviewer #3: Yes

3. Has the statistical analysis been performed appropriately and rigorously? 

Reviewer #2: (No Response)

Reviewer #3: Yes

4. Have the authors made all data underlying the findings in their manuscript fully available?

Reviewer #2: (No Response)

Reviewer #3: Yes

5. Is the manuscript presented in an intelligible fashion and written in standard English?

Reviewer #2: (No Response)

Reviewer #3: Yes

6. Review Comments to the Author

Reviewer #2: (No Response)

Reviewer #3: 1. All comments have been addressed.

2.The manuscript is technically sound, and the data do support the conclusions.

3.The statistical analysis has been performed appropriately and rigorously.

4.The authors have made all data underlying the findings in their manuscript fully available.

5.The manuscript is presented in an intelligible fashion and written in standard English.

7. PLOS authors have the option to publish the peer review history of their article (what does this mean? ). If published, this will include your full peer review and any attached files.

**Do you want your identity to be public for this peer review?** For information about this choice, including consent withdrawal, please see our Privacy Policy .

Reviewer #2: No

Reviewer #3: **Yes: ** Kassa Demissie Abdi (PhD)

---

## [Editor Report · Acceptance letter]

PONE-D-23-16906R2

PLOS ONE

Dear Dr. Newton-Clarke,

I'm pleased to inform you that your manuscript has been deemed suitable for publication in PLOS ONE. Congratulations! Your manuscript is now being handed over to our production team.

Kind regards,

on behalf of

Dr. PLOS Manuscript Reassignment

Staff Editor

PLOS ONE